# Influence of Nutritional Ketosis Achieved through Various Methods on Plasma Concentrations of Brain Derived Neurotropic Factor

**DOI:** 10.3390/brainsci12091143

**Published:** 2022-08-27

**Authors:** Madison L. Kackley, Alex Buga, Chris D. Crabtree, Teryn N. Sapper, Craig A. McElroy, Brian C. Focht, William J. Kraemer, Jeff S. Volek

**Affiliations:** 1Department of Kinesiology, The Ohio State University, Columbus, OH 43210, USA; 2Department of Medicinal Chemistry and Pharmacognosy, 06 The Ohio State University, Columbus, OH 43210, USA

**Keywords:** BDNF, ketogenic diet, BHB, BHB salts, exercise, weight loss

## Abstract

Brain-Derived Neurotropic Factor (BDNF) expression is decreased in conditions associated with cognitive decline as well as metabolic diseases. One potential strategy to improve metabolic health and elevate BDNF is by increasing circulating ketones. Beta-Hydroxybutyrate (BHB) stimulates BDNF expression, but the association of circulating BHB and plasma BDNF in humans has not been widely studied. Here, we present results from three studies that evaluated how various methods of inducing ketosis influenced plasma BDNF in humans. Study 1 determined BDNF responses to a single bout of high-intensity cycling after ingestion of a dose of ketone salts in a group of healthy adults who were habitually consuming either a mixed diet or a ketogenic diet. Study 2 compared how a ketogenic diet versus a mixed diet impacts BDNF levels during a 12-week resistance training program in healthy adults. Study 3 examined the effects of a controlled hypocaloric ketogenic diet, with and without daily use of a ketone-salt, on BDNF levels in overweight/obese adults. We found that (1) fasting plasma BDNF concentrations were lower in keto-adapted versus non keto-adapted individuals, (2) intense cycling exercise was a strong stimulus to rapidly increase plasma BDNF independent of ketosis, and (3) clinically significant weight loss was a strong stimulus to decrease fasting plasma BDNF independent of diet composition or level of ketosis. These results highlight the plasticity of plasma BDNF in response to lifestyle factors but does not support a strong association with temporally matched BHB concentrations.

## 1. Introduction

Cognitive decline occurs at a higher rate in advancing age, but the rate of decay is accelerated/enhanced by poor-metabolic health (i.e., obesity, metabolic syndrome, diabetes) and progressive neurological disease like Parkinson’s Disease (PD), Huntington’s Disease (HD), or Alzheimer’s Disease (AD). A key component of brain plasticity is brain-derived neurotropic factor (BDNF), a neuronal protein that promotes the survival of neurons through growth and maturation. Expression of BDNF can be negatively affected by poor metabolic health [1].

One mechanism for cognitive decline is reduced ability to utilize glucose for brain energy metabolism [2] and decreased brain network stability starting at age 47 [3], which may be circumvented by providing an alternative energy source such as ketones. A novel strategy to naturally improve metabolic health is to increase circulating ketones, namely beta-hydroxybutyrate (BHB), achievable though a ketogenic diet (KD) and/or exogenous ketones raising plasma ketone concentration levels into a range of nutritional ketosis (i.e., 0.5 to 4.0 mM) [4,5]. Many studies have demonstrated improved cognitive performance in various neurological disorders using ketogenic therapies such as BHB infusion [6], ketone ester (KE) ingestion [3,7], and a KD [8,9].

Prior studies in both rats and humans reported a positive correlation between blood BHB and BDNF concentrations, suggesting that ketogenic interventions may improve cognition in part through a BDNF-mediated mechanism [10,11]. Although human studies are limited, this effect appears largely dependent on circulating BHB concentrations [12,13,14]. Besides the KD, circulating ketones can also be increased through exogenous ketone supplementation, which raise BHB rapidly after ingestion [15,16]. Exercise transiently elevates ketone concentrations, which may contribute to increased BDNF during the recovery period [17,18]. The BDNF response to exercise may help combat mental health disorders like depression and anxiety in both human and mouse models [17,19,20]. In humans, increasing aerobic exercise reduces hippocampal atrophy by an increase in BDNF and other growth factors [21]. Plasma BDNF has been previously used in training studies as a surrogate biomarker of brain BDNF production during exercise [18] and circulating plasma BDNF concentrations have been shown to increase after both acute and long-term resistance training implying a possible mechanistic role of ketones and BDNF in exercise training related improvements in neurocognitive performance [22].

Considering a decrease in BDNF concentration is a cardinal feature of cognitive decline and the known effect of BHB to stimulate BDNF expression, relatively little research has been done exploring how ketogenic interventions influence plasma BDNF in humans. Both exogenous ketone esters and chronic KD consumption have been shown to increase BDNF [12,13], but no studies have examined the effects of ketone salt formulations or the impact of combined KD and exogenous ketone interventions. Moreover, there is little known on the impact of exercise and training status on the association between ketones and BDNF. In order to further explore the associations of circulating BHB and BDNF, we analyzed BDNF concentrations in archived plasma samples from three human studies that involved different methods of achieving nutritional ketosis [15,23,24].

## 2. Materials and Methods

### 2.1. Study 1: Ketone Salt Supplementation with Acute Exercise

This project (Ketone Supplement Study or KSUPP) explored how acute ingestion of a ketone salt formulation prior to high-intensity exercise influenced the metabolic response to exercise and performance in a group of healthy recreationally trained adults [15]. In brief, this study tested the effects of a pre-workout supplement containing BHB salts, caffeine, and amino acids (KCA) in recreationally active adults habitually consuming a mixed diet (Keto-Naïve; *n* = 12) or a KD (Keto-Adapted; *n* = 12). In a randomized and balanced manner, subjects consumed either the KCA consisting of 7 g BHB (72% R-BHB and 28% S-BHB) with 100 mg of caffeine, leucine and taurine or water (control condition) 15 min prior to performing a staged cycle ergometer time to exhaustion test followed immediately by a 30 s Wingate test. BDNF concentrations from plasma samples were analyzed via a colorimetric assay at baseline, immediately post exercise, and after 30 min of recovery. Additional details on the testing day protocol have been previously published [15]. Study characteristics can be compared in Table 1. 

### 2.2. Study 2: Prolonged Ketogenic Diet and Resistance Training

This project compared how a KD versus a mixed diet impacts metabolic and performance outcomes during a 12-week resistance training program. This study is an extension of a larger project called Tactical Athletes in Nutritional Ketosis (TANK) designed to evaluate the feasibility, metabolic, and performance responses of an extended duration KD. Healthy adults (*n* = 29) from various military branches participated in a supervised 12-week progressive resistance training program. In brief, fifteen participants self-selected to an ad libitum KD guided by daily measures of capillary blood ketones and fourteen continued their normal mixed diet (MD). BDNF from archived frozen plasma samples were analyzed at baseline and post intervention timepoints. Detailed methods for testing protocol and training have been previously reported [24,25]. Study characteristics can be compared in Table 1. 

### 2.3. Study 3: Ketogenic Diet, with and without Ketone Salts, during Weight Loss

This project, Strategies to Augment Ketosis (STAK), explored the effects of a controlled feeding hypocaloric KD, with and without daily use of a ketone-salt, on BDNF levels in overweight/obese adults [23,26]. Participants were provided a precisely defined hypocaloric KD (~75% of energy expenditure) for 6 weeks. In a double-blind manner, subjects were randomly assigned to receive BHB-salt (KD + KS; *n* = 12) or placebo (KD + PL; *n* = 13). A matched comparison group (*n* = 12) was separately assigned to an isoenergetic/isonitrogenous low-fat diet (LFD). BDNF from archived frozen plasma samples were analyzed at baseline, 2-, 4-, and 6-weeks of the ketogenic intervention. Extensive methods for this study were reported previously [23,26]. Study characteristics can be compared in Table 1. 

**Table 1 brainsci-12-01143-t001:** Study Characteristics.

Methods	Experiment		
	Study 1	Study 2	Study 3
Sample Size	12 M/12 F	25 M/4 F	19 M/18 F
Duration	Cross-sectional	12 weeks	6 weeks
Structured Exercise	Yes	Yes	No
Controlled Feeding	No	No	Yes
Weight Loss	No	KD only	Yes
Ketone Supplements	Yes	No	Yes
Cross-over	Yes	No	No
Randomized	Yes	No	Yes
Double Blind	Yes	No	Yes
Control Group	Yes	Yes	Yes

### 2.4. Blood BHB and Plasma BDNF Analysis

Blood BHB was determined in capillary blood using reagent strips that were inserted into a monitoring device (Abbott FreeStyle^®^, Columbus, OH, USA), and recorded at the time point of analysis. Plasma BDNF was analyzed using an enzyme linked immunosorbent assay (Eagle Biosciences, Lot #101,027, Nashua, NH, USA). Each sample was removed from a −80 °C freezer and then allowed to thaw to room temperature. Each sample was vortexed and then added to a sample dilution buffer that was a ratio of 5% sample and 95% dilution as suggested by the manufacturer. The color development was stopped by the addition of acid and the intensity of the color was measured at a wavelength of 450 ± 2 nm in a SynergyHI microplate reader (Winooski, VT, USA). Samples were analyzed in duplicate and averaged. The concentration of BDNF in the sample was then determined by comparing the Optical Density of samples to the standard curve. The intra-assay coefficient of variance was 7.8%.

### 2.5. Statistical Analysis

The primary analyses explored the association between BHB and BDNF. Data were analyzed using SPSS version 25 (IBM, Armonk, NY, USA). Main effects and interactions were determined using repeated measures Analysis of Variance (RM ANOVA). Prior to analysis we validated the RM ANOVA using Mauchly’s sphericity test. If sphericity was violated (*p* < 0.05), Greenhouse-Geiser corrections were applied. Significant post hoc main effects and interactions were evaluated with Fischer Least Significant Difference due to the exploratory design of the study. Effects sizes (Cohen’s *d*) were calculated for each main outcome variable (0.2 = weak effect; 0.5 = moderate effect; 0.8 = large effect) using pre-post data whenever appropriate. The two-tail alpha level cut-off for significance was set a priori at *p* < 0.05.

Study 1 was a cross-over design with supplement (KS vs. PL) and diet (keto-adapted vs. keto-naïve) as between-subject factors, and time as a within-subject factor. The primary goal was to determine the main effects between BHB and BDNF change induced by diet alone. The secondary goal was to determine if KS further increases BDNF proportional to BHB augmentation post-supplement ingestion. Study 2 consisted of a 2-way repeated measures ANOVA with diet as a between-subjects factor, and time as a within-subjects factor to assess the pre-post changes in BDNF during a 12-week resistance training program on randomized and balanced diets (KD or LFD). Study 3 required a 3 (group) × 4 (time) ANOVA to evaluate bi-weekly BDNF changes across KD + KS and KD + PL over six-weeks and compare the results to the placebo-controlled LFD arm.

Correlations between change in BHB and BDNF were determined in each study, and associations were also explored with glucose (GLU), insulin, body weight (BW) and body fat percentage (BF%).

## 3. Results

For all three studies, BDNF means, SEMS, Effect Sizes, and *p*-values for the ANOVA main effects are shown in Table 2. Additional details of each study are as follows.

### 3.1. Study 1

#### 3.1.1. Fasting/Resting BHB and BDNF

Resting capillary blood R-BHB concentrations in the Keto-Adapted group were 2-fold higher than Keto-Naïve subjects (0.70 vs. 0.33 mM). In order to assess if there were differences in fasting/resting BDNF between Keto-Naïve and Keto-Adapted groups, the pre-exercise BDNF concentrations were averaged between the two trials (KS & WT) in each group. A two-tailed *t*-test revealed that the Keto-Naïve group had 13% higher fasting BDNF than the Keto-Adapted group (*p* = 0.03) (Table 2). The Keto-Adapted group showed greater variability than the Keto-Naïve group (range: 175–327 vs. 233–312 pg/dL) (Appendix A).

#### 3.1.2. Post-Exercise BHB and BDNF

After ingestion of the KS, R-BHB concentrations were higher at all time points during and after exercise compared to Water (Figure 1A). The highest R-BHB concentrations occurred at IP after KS ingestion. Capillary R-BHB returned to concentrations similar to baseline 30 min into recovery in Keto-Adapted subjects. BHB increased from BL to post exercise in KS conditions only (60% in KA and 45% in KN, respectively), with additive effects in the keto-adapted cohort, followed by a sharp decrease to near baseline values by IP30. KA + WT had a significantly lower R-BHB concentration than all other conditions at IP30 (*p* < 0.05).

Intense exercise resulted in a significant increase in BDNF immediately post-exercise (KA + KS, 64.1%; KA + WT, 51.9%; KN + KS, 9.0%; KN + WT, 80.9%) and 30 min into recovery from pre-intervention measurements (*p* < 0.001) (Figure 1B). BDNF at 30 min into recovery showed an increase in all groups except KN +WT from IP concentrations (KA + KS, 7.3%, KA + WT, 8.0%, KN + KS, 25%; KN + WT, −15.2%). BDNF 30 min after exercise remained higher than pre-exercise concentrations in all conditions and groups. All participants showed increased concentrations of BDNF immediately post exercise except for two individuals from the KN + KS group and one individual from the KS + WT group. The KN + KS condition was associated with lower BDNF than KA + KS (*p* < 0.001) and KN + WT (*p* = 0.007), but not KA + WT (*p* = 0.214).

#### 3.1.3. Metabolic and Hormonal Reponses

In both groups and trials, capillary glucose peaked immediately post-exercise, gradually returning to baseline levels during recovery. Plasma insulin remained stable in response to the supplement ingestions and exercise in Keto-Adapted subjects, whereas responses were more variable in Keto-Naïve subjects. 4 of the 12 individuals in the Keto-Naïve group had HOMA-IR values >1.0 compared to none in the Keto-Adapted group.

### 3.2. Study 2

#### 3.2.1. BHB and BDNF

Subjects in the KD group recorded glucose and ketone values 97% of days during the intervention. Mean capillary blood BHB was 1.2 ± 0.4 mM (range: 0.9 to 1.8 mM) in KD participants, indicating each subject was in nutritional ketosis. On average participants reached nutritional ketosis (BHB ≥ 0.5 mM) in 2.9 ± 2.0 days and reported two consecutive days of nutritional ketosis 3.5 ± 2.0 days after beginning the KD [24]. Capillary BHB increased from baseline (pre) to week 12 (post) via dependent *t*-test (*p* < 0.001) (Figure 2A).

There was no difference in BDNF concentrations between the two groups at baseline. Mean BDNF increased modestly from baseline, albeit non-significantly, with no significant differences between diets (Figure 2A). There was a high degree of variability in the BDNF response with six individuals from each group showing a decrease and seven KD and 8 MD individuals showing an increase in BDNF concentration post intervention (Figure 2B).

#### 3.2.2. Metabolic and Hormonal Responses

Glucose remained unchanged post-intervention in the KD (79.1 ± 10.7 to 74.8 ± 6.1 mg/dL) and MD (75.8 ± 5.9 to 74.8 ± 8.7 mg/dL) groups. Insulin was lower postintervention (*p* = 0.005), and the decrease was of greater magnitude in the KD vs. MD group (10.3 ± 4.9 to 5.7 ± 2.4 vs. 8.1 ± 4.6 to 7.3 ± 2.8 µIU/mL; *p* = 0.035). Insulin sensitivity improved in the KD group (2.1 ± 1.2 to 1.1 ± 0.5) and remained unchanged in the MD group (1.5 ± 0.9 to 1.4 ± 0.5) (*p* = 0.042) [24]. Body mass decreased in KD vs. MD (−7.7 vs. 0.1 kg; *p* < 0.001) from baseline to post intervention. All 15 subjects in the KD group lost greater mass than any participant in the MD group [24].

### 3.3. Study 3

#### 3.3.1. BHB and BDNF

In Both KD Groups, fasted capillary BHB increased progressively to concentrations above 1.0 mmol/L by Week 2 and stayed elevated throughout the six-week intervention (Figure 3A). Average daily fasting R-BHB over 42 days was 25% higher in KD + KS (1.28 mM) than KD + PL (1.02 mM). This difference was most prominent during the first 2 weeks.

Plasma BDNF concentrations decreased significantly from baseline starting at WK2 and then continued to decrease through WK4 and WK6 (pre: 617 ± 19 pg/dL to post: 370 ± 33, respectively) (Figure 3B). There was no significant difference between groups at any time point. Post intervention BDNF values decreased in the KD + KS group by 53.4%, KD + PL by 65.5% and in the LFD group by 83.4%. One participant in the KD + PL group increased 1.2% in BDNF concentrations from pre to post, and interestingly this person had one of the lowest BDNF values at baseline.

#### 3.3.2. Metabolic and Hormonal Responses

Fasting capillary glucose at WK6 was decreased to a greater extent in KD + PL (−15 mg/dL) > KD + KS (−7 mg/dL) > LFD (+1 mg/dL). An interaction effect was found in the KD + PL, an effect that was observed at WK2 and was maintained at WK6 (−8 mg/dL). There was no significant glucose lowering effect in LFD. Insulin levels and insulin resistance, calculated from fasting glucose and insulin using the homeostatic model, decreased at WK2 and stayed lower at WK4 and WK6 in all groups (*p* < 0.001).

### 3.4. Study 1 vs. Study 2 vs. Study 3

When comparing the main effect size of each study, Study 3 showed the strongest difference compared to the other two studies (Table 3). The change in BDNF was not correlated with BHB in any study. There were two significant interactions in Study 3. A positive correlation in the KD + PL group between BDNF and glucose concentration was detected (Appendix A), while a negative correlation was found in the LFD group with plasma insulin and BDNF concentrations Appendix A of BDNF main effects and interactions are shown in Appendix A. No correlations were found regarding weight loss or body fat percentage and BDNF concentration (Appendix A).

## 4. Discussion

Contrary to our hypothesis, the findings from three separate studies provide little support for our hypothesis that higher blood BHB in healthy adults consuming either a KD and/or ketone salts would be associated with increased plasma BDNF. We did however make several important discoveries. In Study 1 it was demonstrated that fasting/resting BDNF concentrations were lower in keto-adapted than non-keto-adapted individuals. Plasma BDNF increased after acute exercise regardless of starting ketone levels. Blood samples were taken within 2 min of the completion of the exercise protocol, thus capturing the immediate BDNF response to intense exercise. However, in Study 2 there was no increase in fasting, resting BDNF after a 12-week resistance training regimen, implying that plasma BDNF may be transiently responsive to exercise but does not remain constitutively elevated. Second, in Study 3 it was demonstrated that weight loss achieved over 6-weeks was a robust stimulus to uniformly decrease plasma BDNF regardless of the macronutrient distribution of the diet or level of ketosis. Thus, circulating BDNF levels can be modulated in healthy adults, but unlike the results from many animal studies [11,27] the results from our work do not support a major effect of nutritional ketosis achieved through a low-carbohydrate eating pattern or exogenous ketone consumption on plasma BDNF in healthy adults.

### 4.1. Exercise and BDNF

The increase in BDNF after intense cycling exercise in Study 1 was not unexpected as previous studies have indicated an approximate 50% increase in BDNF after a maximal cycling protocol [28]. While acute exercise consistently increases BDNF, the response to long term training has shown either an increase [3,22,29,30] or no change in resting BDNF [31]. We demonstrated that a 12-week resistance training intervention resulted in no change in BDNF concentrations in the context of either a KD or MD. The lack of BDNF response may be related to training status. Our subjects were enlisted ROTC cadets and cadre regularly participating in scheduled physical activity whereas studies in untrained populations [22,32] demonstrated an increase in resting BDNF following chronic resistance training protocols.

### 4.2. Ketones and BDNF

Ingestion of a ketone-salt that modestly increased blood BHB prior to a maximal acute exercise protocol did not further increase BDNF concentrations. It is possible that a higher BHB level, achievable for example with higher dose ketone esters, could create a significant change in BDNF after acute anaerobic exercise as it has been published that salts elicit a smaller BHB response than ketone esters [16,33]. Two previous studies found an increase in BDNF after adaptation of ketosis (by both KD and acute KE ingestion), but one trial where participants were fed a mixed diet with daily ingestion of 12 g of KE thrice daily found no difference in BDNF concentrations after the 14 day dietary intervention [12,13,14].

The KN + KS, whose baseline values were significantly higher than the other baseline trial values, had the least amount of increase at IP by 9.0%. It should be noted that although not significant, the KN + WT was the only group to see a drop in BDNF concentrations at the 30 min post exercise mark (IP 426 ± 22 pg/dL, IP30 370 ± 26 pg/dL; Mean ± SD). This would be the only group in this crossover trial to not be exposed to elevated ketones. Brunelli et al. [28] reported a significant drop during a recovery period in the BDNF concentrations of 10 healthy men who performed a cycling incremental test to exhaustion, much like the one in this protocol. It is possible that the presence of BHB may have deterred BDNF from dropping in the recovery period. In comparison to KN + WT, recovery levels of BDNF after acute exercise were significantly higher in the KA + KS group but not significantly different than the KA + WT and KN + KS groups. Further research is needed to resolve whether heightened levels of ketosis are needed to elicit a larger response in BDNF concentrations during recovery.

With the exception of the unexpected lower resting BDNF in keto-adapted individuals (Study 1), no differences were seen in BDNF responses between a KD and non-ketogenic diet in Studies 2 and 3. Thus, there does not appear to be a strong effect of chronic elevations in ketones on plasma BDNF. None of the participants in the three studies were in a cognitive deficit, which may inhibit the potential cognitive effects that may be seen in people with schizophrenia, autism and depression [34,35,36]. This lack of change across all methods of induced ketosis, requires further research on the level of ketosis needed to achieve an increase in BDNF levels for populations with and without cognitive decline.

### 4.3. Weight Loss and BDNF

A 6-week hypocaloric diet, irrespective of macronutrient composition, showed a highly uniform and progressive decrease in plasma concentrations of BDNF. The post intervention concentrations of BDNF were markedly decreased in all of the groups (KD + KS group by 53.4%, KD + PL by 65.5% and in the LFD group by 83.4%) but this did not correlate with weight loss. All participants in the 3 groups demonstrated a decrease in BDNF except for one participant in the KD + PL group with a particularly low starting BDNF who increased 1.2% from pre to post. The baseline values of these participants were almost two-fold higher than any of the other cohorts examined in this study and the post-weight loss values are comparable to those reported in other studies. Elevated BDNF levels have been shown in populations with obesity or metabolic syndrome [37,38]. There has been a reported positive correlation between the number of metabolic risk factors and BDNF [38] because long term exposure to metabolic risk factors lead to the central nervous system’s (CNS) overproduction of BDNF, due to its role in regulating energy management. Although the relationship between BDNF and obesity has yet to be defined, obesity is associated with increased risk of neurodegenerative disease [39]. By the time a person is diagnosed with a metabolically influenced cognitive disease, previously elevated BDNF levels have dropped significantly [40,41,42]. It is possible that this population saw a reduction in BDNF levels after weight loss due to the regulation of potential metabolic risk factors such as body fat and the decreased need for the CNS to compensate for metabolic imbalance. BDNF deficient populations, such as those who have cognitive impairment, benefit from a hypocaloric diet to help BDNF regulation [43]. None of the participants had neurological problems that would necessitate a heightened need for increased BDNF.

In light of the highly significant decrease in BDNF after weight loss in Study 2, it is somewhat perplexing that the KD group in Study 3 demonstrated no change in BDNF. This discrepancy may be explained by metabolic health status with participants in Study 3 being healthier and having lower baseline (pre-intervention) BDNF than Study 2. Thus, the CNS did not need to excrete higher levels of BDNF to resolve impaired metabolism. Another possible reason BDNF was not decreased after significant weight loss in Study 3 may be due to the structured exercise program, which acted to counteract the expected drop in BDNF upon weight loss.

### 4.4. Exploratory Effects of Different Markers of Health on BDNF

The adult brain is dynamic, meaning that it can be changed by internal factors like neurological homeostasis but also external factors such as metabolic health [44]. Metabolism imbalance has been shown to be associated with decreased cognitive function [45]. GLU and insulin are among two of the most prominent markers that are used in diagnosis of insulin sensitively and energy management [46,47].There is a pronounced role of BDNF in the regulation of feeding and energy metabolism, therefore, the idea that metabolic imbalances can be mediated with dietary interventions can be appreciated. Increased BDNF concentrations in persons with Type II Diabetes illicit increased metabolic effects such as the upregulations of neurotrophin receptor tyrosine kinase (TrkB) and cAMP-response element binding protein (CREB) pathways. These pathways can reduce levels of GLU and insulin in peripheral tissues, thereby improving insulin sensitivity [48]. Both markers can be strong indicators of the capacity to utilize fatty acids in relation to insulin resistance and obesity.

Exploratory analyses of GLU and insulin were analyzed to explain the potential effects of changes in metabolism, specifically fat, on BDNF. Although both KD groups had a marked decrease in fasting glucose concentrations, a positive correlation was detected in the KD + PL group between BDNF and glucose concentrations. KDs have historically been used to help regulate insulin sensitivity and ultimately, energy homeostasis, which hypothetically would allow for better neurological regulation [45], potentially leading to this positive relationship. However, the same relationship may have been blunted in the KD + KS group due to the decrease in lipolysis associated with exogenous ketone supplementation [49], which may have altered the endogenous metabolic change that is typically seen with insulin and GLU responses after a KD intervention. In a hypocaloric LFD, a negative correlation between plasma level insulin concentrations and BDNF was detected. This group would not have seen the metabolic effects associated with fat oxidation that the KD did, thus, insulin may not have been well regulated, even after weight loss, leading to a decrease in BDNF [41,50]. Thus, further work is needed to explore the effects of changes in metabolism on BDNF due to both diet variation and weight loss.

## 5. Conclusions

In summary, the findings of this investigation reveal that resting plasma BDNF is lower in keto-adapted than non-keto-adapted individuals and that BDNF is highly responsive to intense exercise protocols regardless of prior diet. A single bout of intense cycling increased plasma level BDNF concentrations, but chronic (12-week) resistance training did not significantly affect resting levels of BDNF in healthy adults consuming either a KD or a MD. Lastly, it was revealed that a hypocaloric diet resulted in a consistent and marked decrease in BDNF irrespective of diet composition and ketosis in overweight and obese populations. Further work is needed to determine the transient nature of plasma BDNF responses to different types of exercise, the impact of volume or frequency of exercise training on resting BDNF concentrations, and the role of more potent ketogenic interventions such as the KD and exogenous ketones on these responses.

## Figures and Tables

**Figure 1 brainsci-12-01143-f001:**
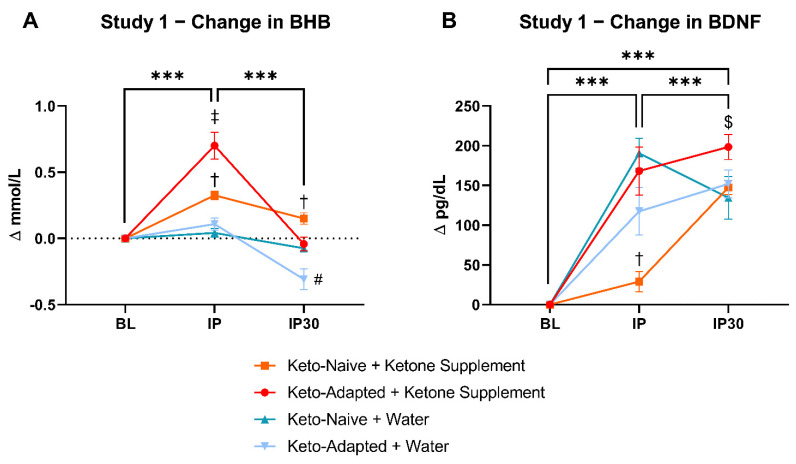
Study 1 BHB and BDNF Changes. Two unique groups (Keto-adapted vs. Keto-naïve) were enrolled in parallel and crossed over to receive either a KS or WT before exercise. Capillary BHB was normalized to baseline and analyzed for covariance (ANCOVA) using a 4 (condition) × 3 (time) model. BHB increased acutely from BL to post exercise in KS conditions only, with additive effects in the keto-adapted cohort, followed by a sharp decrease to baseline values by IP30 (**A**). Plasma BDNF was normalized to baseline and analyzed for covariance (ANCOVA) using a 4 (condition) × 3 (time) model. BDNF increased acutely from baseline to immediately post-exercise (Δ = 126 ± 12 pg/dL; +54%; *p* < 0.001) and 30 min thereafter (Δ = 158 ± 12 pg/dL; +63%; *p* < 0.001). The KN + KS condition stimulated less BDNF than KA + KS (*p* < 0.001) and KN + WT (*p* = 0.007), but not KA + WT (*p* = 0.214) (**B**). Time effects: *** *p* < 0.001. † = KN + KS different than all conditions at IP (*p* < 0.05). ‡ = KA + KS different than all conditions at IP (*p* < 0.05). # = KA + KS different than all conditions at IP30 (*p* < 0.05). $ = KA + KS different than KN + WT at IP30 (*p* < 0.05). BL, baseline; IP, immediately post-exercise; IP-30, immediately post 30 min after exercise cessation.

**Figure 2 brainsci-12-01143-f002:**
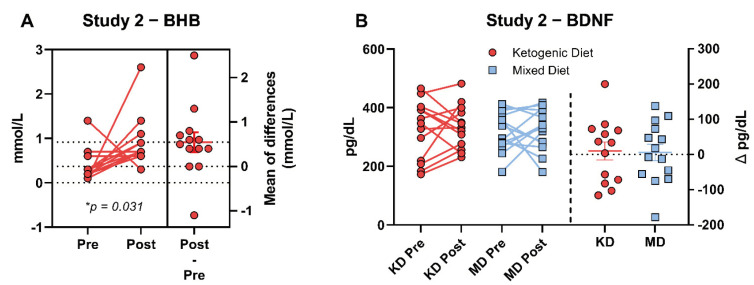
Study 2 BHB and BDNF Changes. Pre-post capillary BHB was evaluated via dependent *t*-test in the KD group (**A**). Change from pre to post was significantly increased (*p* < 0.05). Fasted plasma BDNF was calculated using a 2 (diet) × 2 (time) analysis of variance (ANOVA). BHB increased over the 12-weeks (Δ = 0.55 ± 0.2 mmol/L; +249%; *p* = 0.031) within the range of nutritional ketosis. There was no significant change in BDNF in either group (**B**).

**Figure 3 brainsci-12-01143-f003:**
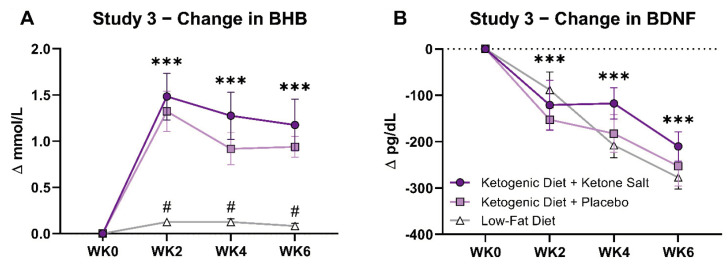
Study 3 BHB and BDNF Changes. In both KD groups, fasted capillary BHB increased progressively to concentrations above 1.0 mmol/L by WK2 and stayed elevated throughout the 6-week intervention. Fasting BHB was higher in KD + KS at WK1 and WK2 but was similar to KD + PL thereafter (**A**). Plasma BDNF was calculated using a 3 (diet) × 4 (time) repeated measure analysis of variance (ANOVA). Plasma concentrations of BDNF decreased significantly from baseline starting at WK2 (Δ = −121 ± 20 pg/dL; −17%; *p* < 0.001), then continued to decrease through WK4 (Δ = −170 ± 20 pg/dL; −26%; *p* < 0.001) and WK6 (Δ = −247 ± 20 pg/dL; −39%; *p* < 0.001) (**B**). Time effects: *** *p* < 0.001 from WK0. # = different than KD (*p* < 0.05). WK0, week 0; WK2, week 2; WK4, week 4; WK6, week 6.

**Table 2 brainsci-12-01143-t002:** BDNF Main Effects and Interactions.

Study	Condition	Timepoints				Change	Effect Size	RM ANOVA (*p*-Values)		
		Pre	Midpoints		Post	Post–Pre	*d*	Condition	Time	Interaction
Study 1	KN + KS	322 ± 10 ^#^	351 ± 12		469 ± 13	147 ± 5	3.7	**0.011**	**<0.001**	**<0.001**
	KA + KS	257 ± 15	423 ± 34		456 ± 18	198 ± 7	3.5			
	KN + WT	235 ± 6	426 ± 22		370 ± 26	134 ± 8	2.0			
	KA + WT	234 ± 17	355 ± 27		386 ± 16	152 ± 7	2.7			
Study 2	KD + KS	603 ± 31	482 ± 42	486 ± 27	393 ± 19	−210 ± 10	2.4	0.54	**<0.001**	0.15
	KD + PL	639 ± 27	487 ± 22	457 ± 30	386 ± 27	−253 ± 11	2.6			
	LFD	610 ± 28	522 ± 25	402 ± 23	332 ± 28	−277 ± 11	2.9			
Study 3	KD	325 ± 28			335 ± 20	10 ± 10	0.1	0.76	0.64	0.91
	MD	319 ± 18			325 ± 19	6 ± 7	0.1			

All values reported as mean ± SEM (pg/dL). Significant effects (*p* < 0.05) are highlighted in bold face. Effect size cut-off values: 0.2 = weak effect; 0.5 = moderate effect; 0.8 = strong effect size. ^#^ = significantly higher BDNF concentration compared to the other conditions in Study 1. KN = Keto-Naïve +; KA = Keto-Adapted; KS= Ketone Salt; WTR = Water; KD= Ketogenic Diet; LFD= Low Fat Diet PL= Placebo; MD= Mixed Diet.

**Table 3 brainsci-12-01143-t003:** BDNF Effect Size Matrix (Cohen’s *d*).

Study	Condition	Study 1				Study 2		Study 3		
		KN + KS	KA + KS	KN + WT	KA + WT	KD	MD	KD + KS	KD + PL	LFD
**Study 1**	KN + KS									
	KA + KS	0.9								
	KN + WT	0.2	0.8							
	KA + WT	0.1	0.8	0.2						
**Study 2**	KD	2.3	2.8	1.5	2.2					
	MD	2.4	2.9	1.6	2.3	0.1				
**Study 3**	KD + KS	1.1	0.2	1.0	1.0	2.9	3.0			
	KD + PL	1.4	0.7	1.3	1.3	2.8	2.9	0.5		
	LFD	1.7	1.0	1.5	1.6	3.1	3.2	0.8	0.3	

To compare between-studies BDNF effects we calculated the pre-post BDNF change within each study condition, followed by a series of Cohen’s *d* tests examining the magnitude differences. To visualize the effect sizes, we color-coded the matrix with a light and dark green gradient to denote weak and strong effect sizes, respectively. Effect size cut-off values: 0.2 = weak effect; 0.5 = moderate effect; 0.8 = strong effect size.

## Data Availability

The data analyzed and presented in this manuscript can be provided upon reasonable request.

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
