# Peer review of "Influence of Nutritional Ketosis Achieved through Various Methods on Plasma Concentrations of Brain Derived Neurotropic Factor"

_brainsci, 2022, doi:10.3390/brainsci12091143_

Round 1
Reviewer 1 Report
This is a colossal amount of work. Aside from a few typos which will be picked up by the copy editor, I suggest accept without revision.
Author Response
We appreciate the reviewer's time and support. Thank you for your comment.
Reviewer 2 Report
The manuscript “Influence of Nutritional Ketosis Achieved through Various Methods on Plasma Concentrations of Brain Derived Neurotropic Factor” comprises three independent human trials in which BDNF levels are measured following different interventions related with ketogenesis. The authors determine that interventions that modulate ketogenesis levels associate with plasma BDNF changes, but do not observe a strong association with concentrations of the ketoacid β-hydroxybutyrate.
The methods and approaches for this preliminary study are adequate, and the work provides relevant information regarding the relationship between ketogenesis and BDNF levels in human patients, with some findings being unsurprising, and others unexpected and interesting. The introduction adequately summarizes prior background, and the discussion provides interesting references and ideas. My main concern with the work is related with the structure of the manuscript itself, as it makes it quite difficult to understand each study and respective analysis, and this should be more clear and understandable for the readership. Regarding this issue, my suggestion is the following:
* The methodology of each individual study should be followed by the results section of that particular study, otherwise it is very hard to maintain follow-up. This Method-Result-Method-Result-Method-Result structure is not the conventional one, but it would largely optimize the manuscript. The “Blood BHB and Plasma BDNF Analysis” and “Statistical Analysis” can be kept in a general methods section.
* The used abbreviations are very similar, and they must be clearly defined in each table and figure. For example, there is no definition of any abbreviation in the caption of Table 2. Additionally, I would suggest to not use “WT” as an abbreviation for “Water”, as this is easily confused with the designation of “Wild-Type”.
* In study 1, the use of “BL”, “IP” and “IP-30” is not instinctive, I would suggest to not use abbreviations, changing to, for example, “Baseline”, “Immediately Post Exercise” and “30 Min After Exercise”.
Additional issues:
* In Table 2, authors use (appropriately) a RM-ANOVA model to assess for the differences between timepoints. I suggest they should include a supplementary table showing the post-hoc analysis between each time-point comparison (base vs pre, base vs post, pre vs post) for each of the groups in each of the studies.
* The authors should present the data (graph or distribution, in supplementary) of the levels of R-BHB in Keto-Naïve vs Keto-Adapted subjects in Study 1, as well as all data regarding the 3.1.3 section.
* In study 1, the authors should further discuss why the KN+KS group had a much different baseline BDNF when compared to KN+WT allocated subjects, as these are all similar KN subjects prior to the intervention (random allocation issues?).
* In study 1, the authors should discuss why BHB levels 30 minutes after exercise decrease from baseline in the KA+WT group when compared to KN+WT (ketone consumption in an already more ketone-dependent metabolism in KA subjects?).
* In study 2, what were the BHB levels in the subjects undergoing a regular diet? Was this significantly different from those undergoing KD?
* In the end of the 3.2.1 section there is indication of a “Supplemental Figure” which does not exist, and whose associated data is in Figure 2b, left side (typo?).
* In study 2, have authors carried out the Figure 2b analysis while excluding the 3 individuals undergoing KD which did not have an increase in BHB through the diet (the datapoints in Figure 2a which have a mean difference of BHB of 0 or -1)?
* In section 3.4 it is stated that Study 3 showed the strongest difference compared to the other two studies, while referring to Table 3 – this table does not show differences in BDNF following the different treatments. Furthermore, Study 2 shows the strongest main effect interactions with both Study 1 and Study 3.
* While I find the analysis on Table 3 rather interesting, it is important that the authors describe more in detail its meaning and findings, as the text in section 3.4 is very lacking.
* For the discussion: it is interesting that there is no clear picture when it comes to the relationship between BHB and BDNF, but the overall results seem to suggest that more calorie-restricted diets associate with lower BDNF values, independently of the caloric source (mostly observed in Study 3, where BDNF decreased in both diets). Is BDNF production in this case mostly a marker of caloric intake? How can this relate to what is known on the role of BDNF in metabolism? This is already slightly discussed in the last paragraph of the discussion section, but I think it could be expanded.
* There should be a discussion paragraph stating the several limitations of the study, which are not at all discussed (sample size limitations, attempting to correlate different studies with different conditions, etc.).
I thank the authors for considering my suggestions, and hope they can improve the quality of their work.
Author Response
We thank the reviewer for their comments. This aided us in elevating the integrity of our paper. Our responses can be found in the attached documents.

Reviewer 3 Report
Dear Authors,
I have evaluated the manuscript titled “Influence of Nutritional Ketosis Achieved through Various 2 Methods on Plasma Concentrations of Brain Derived Neuro-3 tropic Factor” for publishing to the Brain Science Journal.
The authors investigated the results of three different studies in which effecting factors on plasma BDNF levels such as intense cycling, 6 weeks, and 12 weeks of resistance training programs, and keto-adapted versus non keto-adapted diets.
I think that this aspect of the study has important and valuable results in terms of examining and evaluating the factors that may be effective on BDNF. My evaluations are below in order to make the studies and their results more understandable at some points.
In the summary, besides BDNF, hormonal and metabolic measurements, which are important in the evaluation of the study, should also be stated.
It is necessary to give more clearly and clearly how many groups are in each study protocol, control groups as naive, water or placebo in each group, and the characteristics of these groups.
In the abbreviations made, it is necessary to clearly and clearly write which group it belongs to. For example, the use of KCA was stated in the 1st study protocol; however, in Table 2, the abbreviation was made as KS.
The number of the 3rd study group given in Table 1 needs to be checked again. In addition, in which study groups the Low-fat diet was used.
After these corrections, it can be accepted for publishing in Brain Science Journal,
Sincerely Yours,
Author Response
We are grateful for the reviewers time and effort in aiding to the quality of this manuscript. Reviewer comments and our responses have been attached in file below.
